# A Pan-RNase Inhibitor Enabling CRISPR-mRNA Platforms for Engineering of Primary Human Monocytes

**DOI:** 10.3390/ijms23179749

**Published:** 2022-08-28

**Authors:** Kanut Laoharawee, Matthew J. Johnson, Walker S. Lahr, Christopher J. Sipe, Evan Kleinboehl, Joseph J. Peterson, Cara-lin Lonetree, Jason B. Bell, Nicholas J. Slipek, Andrew T. Crane, Beau R. Webber, Branden S. Moriarity

**Affiliations:** 1Department of Pediatrics, University of Minnesota, Minneapolis, MN 55455, USA; 2Masonic Cancer Center, University of Minnesota, Minneapolis, MN 55455, USA; 3Center for Genome Engineering, University of Minnesota, Minneapolis, MN 55455, USA

**Keywords:** primary human monocytes, pan-RNase inhibitor, CRISPR-Cas9, base editor, genome engineering

## Abstract

Monocytes and their downstream effectors are critical components of the innate immune system. Monocytes are equipped with chemokine receptors, allowing them to migrate to various tissues, where they can differentiate into macrophage and dendritic cell subsets and participate in tissue homeostasis, infection, autoimmune disease, and cancer. Enabling genome engineering in monocytes and their effector cells will facilitate a myriad of applications for basic and translational research. Here, we demonstrate that CRISPR-Cas9 RNPs can be used for efficient gene knockout in primary human monocytes. In addition, we demonstrate that intracellular RNases are likely responsible for poor and heterogenous mRNA expression as incorporation of pan-RNase inhibitor allows efficient genome engineering following mRNA-based delivery of Cas9 and base editor enzymes. Moreover, we demonstrate that CRISPR-Cas9 combined with an rAAV vector DNA donor template mediates site-specific insertion and expression of a transgene in primary human monocytes. Finally, we demonstrate that SIRPa knock-out monocyte-derived macrophages have enhanced activity against cancer cells, highlighting the potential for application in cellular immunotherapies.

## 1. Introduction

Monocytes are myeloid lineage leukocytes derived from hematopoietic stem cells (HSCs) and make up 8–15% of peripheral blood leukocytes. Monocytes have been broadly classified into three distinct subsets; CD14^++^ CD16^−^ (classical), CD14^+(+)^ CD16^+^ (intermediate) and CD14^+^CD16^++^ (non-classical) [1]. Monocytes have several well-established effector functions, including cytokine secretion and differentiation into downstream effectors, namely monocyte-derived macrophages (MDMs) and monocyte-derived dendritic cells (MDDCs) [2,3,4]. Monocytes and their downstream effectors are armed with a functional diversity of receptors that allow them to execute a variety of immunological defense mechanisms including phagocytosis of pathogens and antigen presentation to T and B cells [5].

Current knowledge of monocyte biology and downstream effectors has been partially ascertained from studying monocyte cell lines or murine monocytes [6,7,8]. In particular, THP-1 cells, an immortalized human monocytic leukemia, has been a valuable tool for investigating the biological function of monocytes in disease [6]. However, there are many limitations associated with THP-1 cells and other monocyte cell lines for the study of basic monocyte biology, such as differences in cytokine expression and effector functions [6,7]. Similarly, mouse models utilizing genetic knockouts of specific genes and adoptive transfer experiments have facilitated numerous discoveries of basic biology relating to human monocytes [8]. However, although there is a level of similarity between the gene expression profiles of mouse and human monocytes, there are also key differences that affect development and functional pathways. Therefore, the ability to study primary human monocytes and not their murine counterparts or immortalized cell lines will enhance our understanding of human monocyte biology.

Monocytes, MDMs, and MDDCs are attractive candidates for adoptive cell therapy for many diseases. They inherently home to many tissues, including heart, lung, liver, spleen, muscle, bone, lymph nodes, skin, kidney, brain, and tumors [9,10,11,12,13]. They also have many distinct effector functions, such as tissue remodeling and repair, self-tolerance, infection eradication, inflammation, and cancer surveillance [5,14,15,16,17,18]. In addition, key immune functions of MDMs and MDDCs include phagocytosis, antibody-dependent cellular phagocytosis (ADCP), antibody-dependent cellular cytotoxicity (ADCC), cytokine secretion, and antigen presentation to regulate innate and adaptive immune responses [1,14,19,20,21]. Critical to cancer immunotherapy, monocytes have been shown to robustly home to and infiltrate tumors where they are influenced by the tumor microenvironment and can differentiate into tumor-associated macrophages (TAMs) that either inhibit or promote tumor progression via various mechanisms [14]. In addition, cancers escape ADCP and ADCC by expressing CD47 on the plasma membrane to bind to SIRPα-expressing TAMs to inhibit these anti-cancer mechanisms [22]. The ability to genetically modify monocytes to enhance anti-tumor activities and suppress tumor promoting activities is thus an attractive concept for monocyte-based immunotherapies.

Recently, CRISPR-Cas9 has enabled genome engineering of many cell types that were once highly intransigent to genetic modification, including primary human immune cells. Over the past several years, numerous publications have reported high efficiency CRISPR-Cas9-based genetic engineering of primary human T cells [23,24,25,26,27], NK cells [28,29,30], and B cells [31,32,33] enabling advances in basic mechanistic research and new avenues to improve cell-based immunotherapies. Conversely, methods for efficient genetic manipulation of monocytes remain limited [34]. Here, we report that CRISPR-Cas9-based ribonucleoproteins (RNPs) can be used for high efficiency genome editing of primary human monocytes. In addition, we demonstrate that the use of an RNase inhibitor enables the deployment of mRNA encoded genome editing tools, including Cas9 nucleases and Cas9 base editors. Moreover, we show that Cas9 nuclease combined with a recombinant adeno-associated viral vector (rAAV) used as a DNA donor vehicle can induce site specific homology-directed repair (HDR) for targeted transgene integration. Finally, we demonstrate that SIRPa knocked-out MDMs can enhance activity against a CD47-expressing cancer cell line.

## 2. Results

### 2.1. Electroporation Allows for Efficient Transfection of Nucleic Acids in Primary Human Monocytes

We tested several electroporation parameters of Neon electroporator system (Table 1) with chemically modified mRNA encoding EGFP to determine the optimal transfection conditions for primary human monocytes. At 24 h post-transfection we assessed cell counts and viability using trypan blue exclusion and determined the transfection efficiency as a frequency of GFP expressing cells using flow cytometry (Appendix A). While we observed similar viability and cell counts across all electroporation protocols (Figure 1A,B, respectively), the frequency of GFP expressing monocytes (GFP+CD14+) was highest with the E5 protocol (1700 volts, 20 milliseconds, 2 pulses) (Figure 1C). Interestingly, we observed high variation of mean fluorescent intensity (MFI) of GFP among donors in all electroporation protocols (Figure 1D). Taken together, these results demonstrate that the E5 protocol is best for transfecting nucleic acids into primary human monocytes and therefore the E5 protocol was utilized for all subsequent experiments described below.

### 2.2. CRISPR-Cas9 RNPs Mediate Efficient Editing of Primary Human Monocytes

In order to test the efficiency of CRISPR-Cas9 mediated editing in monocytes, we implemented chemically modified sgRNAs targeting *PDCD1*, *TP53*, *CCR5*, *CD14*, and *AAVS1*. CRISPR-Cas9 ribonucleoproteins (RNPs) complexes with these chemically modified sgRNAs were then individually electroporated into monocytes and the cells were cultured for 5 days before Sanger sequencing and analysis for non-homologous end joining derived insertions/deletions (indels), using the TIDE algorithm [35]. CRISPR-Cas9 RNP transfection in monocytes induced highly efficient (>75%) and consistent indel formation at all tested loci (Figure 2A,B). These data demonstrate that CRISPR-Cas9 RNPs can be used to efficiently edit the genome of primary human monocytes.

To manipulate a therapeutically relevant gene in monocytes, we targeted the signal regulatory protein alpha (*SIRPA*) gene. SIRPα is expressed on monocytes, MDMs, and MDDCs where it interacts with its ligand CD47, which is expressed on most cells as a “don’t eat me” signal, but is often overexpressed by tumor cells to escape immune checkpoint [36,37]. Systemic administration of antibodies against SIRPa/CD47 has been shown to enhance macrophage phagocytosis of CD47-expressing tumors in mice [36,38,39]; however, toxicity due to systemic administration of antibodies has been reported [36]. Therefore, a CRISPR-Cas9 *SIRPA* knockout strategy may reduce potential systemic toxicity associated with antibodies. Monocytes were thus electroporated with CRISPR-Cas9 RNPs targeting the *SIRPA* locus and cultured for 5 days. Subsequent TIDE analysis demonstrated indel frequencies of approximately 70% at the genomic level which corresponded to a 65% reduction in SIRPα expressing monocytes (Appendix A) in the population as measured by flow cytometry with PE-conjugated anti-SIRPa antibody (clone 602411) (Figure 2C; anti-SIRPa antibody see Appendix A). These data indicate that CRISPR-Cas9 RNP can mediate efficient gene knockout in primary human monocytes that results in protein loss.

### 2.3. mRNA Encoding Cas9 Results in Inefficient Genome Editing in Primary Human Monocytes

Delivery of CRISPR-Cas9-based systems using an mRNA format instead of RNP would allow for more flexibility and broader applications, particularly when specific editing enzyme proteins are not readily producible; for example, as noted with base editors and the prime editing systems [40,41]. To test mRNA-based CRISPR-Cas9 editing, we compared Cas9 protein with Cas9 mRNA for genome editing in primary human monocytes. To this end, monocytes were transfected with either *SIRPA*-targeting or *CD14*-targeting chemically modified sgRNA and chemically modified Cas9 mRNA or as RNP (complexed with Cas9 protein). *SIRPA*-targeted RNP induced~70% indels and *CD14*-targeted RNP induced ~80% indels, whereas sgRNA combined with Cas9 mRNA induced ~10% indels for *SIRPA* and ~35% indels for *CD14* (Figure 2D). Moreover, Cas9 mRNA resulted in either low efficiencies or highly variable indel rates relative to the consistent indel levels observed with Cas9 RNP transfected monocytes (Figure 2D). Interestingly, the variable gene editing observed with Cas9 mRNA is in line with our previous observation of variable MFI with GFP mRNA transfection in monocytes (Figure 1D). Given these results, we hypothesized that there may be an mRNA inhibition or degradation mechanism active in monocytes.

### 2.4. Use of RNase Inhibitor results in More Consistent and Efficient RNA-Based Gene Editing in Primary Human Monocytes

Monocytes are a critical component of the innate immune response and are thus equipped with various intracellular response mechanisms to detect foreign genetic material [42,43]. In particular, previous studies have reported the RNase A superfamily as an innate mechanism to protect against foreign RNAs [44,45,46], and the BLUEPRINT project (www.ebi.ac.uk.gxa, accessed on 30 July 2022) reported that many RNases are highly expressed in human monocytes (Figure 3A). This is in stark contrast to the near absence of RNase A superfamily expression in primary human T cells (Figure 3A) [47]. In addition, RNase Alert assay showed inconsistent but high level of intracellular RNase activities among donors, while we observed consistent and low RNase activities among T cell donors (Figure 3B). Furthermore, RNase activities in the monocyte lysate is significantly higher (~5-fold) than in the T cell lysate (Figure 3B). Moreover, a pan-RNase inhibitor was recently reported to allow higher mRNA expression and enhanced mRNA-based genome engineering in non-human primate hematopoietic cells [48]. Thus, we hypothesized that electroporating heterologous mRNA/sgRNAs into the cytoplasm may activate or be subject to mRNA restriction mechanisms in monocytes, leading to a degradation or inhibition of heterologous mRNA expression. A ‘Protector’ RNase Inhibitor (iRNase, Roche) is a protein derived from rat lung that has pan inhibition of the RNase A superfamily [49] and has been historically used for RNA-related in vitro experiments to prevent RNA degradation [50]. To overcome this issue and allow mRNA delivery of genome editing reagents in monocytes, we deployed iRNase in the electroporation reaction.

We first investigated the effects of iRNase on mRNA expression by varying concentrations of 0 (zero), 1:10, 1:20, 1:50, 1:100 dilutions of iRNase in the electroporation of an EGFP encoding mRNA in primary human monocytes and flow cytometry analysis was performed to measure the expression of GFP as an indicator of mRNA expression at 24 h post-transfection (Appendix A). While we observed a similar frequency of GFP-positive cells in all conditions (Figure 3C), iRNase inclusion significantly increased MFI across all but the 1:100 dilution (Figure 3D). The highest MFI was observed in samples with 1:20 and 1:50 dilutions of iRNase, which were up to 3-fold higher than untreated samples (Figure 3D). These results suggest that RNases expressed by monocytes may be responsible for reduced expression and gene editing observed with *EGFP* and Cas9 mRNA, respectively. In light of these findings, we utilized the iRNase (at a final concentration of 1:50 dilution (10.8 Units)) for all subsequent experiments.

Next, we wanted to investigate whether iRNase could improve Cas9 mRNA mediated genome editing in primary human monocytes. To this end, we targeted *SIRPA*, beta-2-microglobulin (*B2M*) or programmed cell death ligand 1 (PD-L1). B2M is a part of the constant region of major histocompatibility class I (MHC I) that is constitutively expressed on all nucleated cells, while PD-L1 is conditionally expressed in monocytes and MDMs and has been reported to regulate other immune cells in the tumor microenvironment^3^. iRNase was electroporated along with Cas9 mRNA and either *SIRPA*, *B2M* or *CD274 (PD-L1)* sgRNA in monocytes and cultured for an additional 5 days. Through TIDE analysis, we observed consistent and significant increases in indel formation at *SIRPA* (Figure 4A), *B2M* (Figure 4B) and *PD-L1* (Figure 4C) with iRNase compared to no iRNase controls (Figure 4A–C). In addition, through flow cytometry analysis, we observed consistent and significant % protein loss of SIRPa (Figure 4A; SIRPα gating see Appendix A), B2M (Figure 4B; B2M gating see Appendix A) and PD-L1 (Figure 4C; PD-L1 gating see Appendix A) in the samples with iRNase compared to no iRNase controls. In contrast, inclusion of iRNase did not similarly enhance mRNA-based gene editing in primary human T cells. For example, as part of an unrelated study, our lab used mRNA encoding editors to simultaneously knock-out *CD3*, *B2M*, and *PDCD1* in T cells in the presence or absence of iRNase and did not observe enhanced gene editing (Appendix A). These data indicate that the inclusion of iRNase during electroporation improves mRNA-based genome editing in primary human monocytes, but has no effect in human T cells, likely because T cells have low levels of RNases.

### 2.5. iRNase Enables other RNA-Based Genome Engineering Platforms in Primary Human Monocytes

Cas9 nuclease editing induces a double strand break (DSB) at target loci and in some cases, off-target loci, which increases the risk for DSB-associated byproducts including chromosomal rearrangements and translocations. This is particularly problematic in multiplex editing approaches, where two or more targeted DSBs are intentionally created [39]. However, alternative genome editing methods that do not create DSBs, such as DNA base editors, can largely circumvent these issues [27,51]. Liu and colleagues have described two highly active classes of DNA base editors, cytosine base editors (CBEs) and adenine base editors (ABEs) that mediate transition mutations without DSB by converting a C•G into T•A and A•T into G•C, respectively [51,52]. To date, no protein form of base editors are readily or commercially available; therefore, we applied the protective iRNase to facilitate transfection of codon optimized cytidine base editor (coBE4) mRNA to mediate base editing of *B2M*, *PDCD1* (PD-1), or a combination of both. Successful conversions of C to T (C > T) results in the elimination of a splice donor site for each gene, effectively leading to gene knockout (KO) [27]. sgRNAs targeting *B2M* along with coBE4 mRNA and iRNase were transfected into primary human monocytes and cultured for 5 days before analysis. Flow cytometry demonstrated a significant reduction in B2M protein expression relative to unedited monocytes **(**Figure 5A; B2M gating see Appendix A). As PD-1 protein is not commonly expressed in monocytes [53], we did not measure percent reduction of expression of this protein. Sanger sequencing of the respective amplicons were analyzed using EditR [54] and revealed ~70% C > T conversion at the *B2M* target site and ~30% C > T conversion at the *PD1* target site (Figure 5B,C). Targeting both *B2M* and *PD-1* simultaneously resulted in C > T conversion rates of ~80% (Figure 5B) and ~40%, respectively (Figure 5C), while no editing was observed in controls. These results indicate that inclusion of iRNase enables efficient CBE mRNA mediated genome editing for single or multiplex base editing in primary human monocytes.

### 2.6. CRISPR-Cas9 and rAAV Mediated HDR for Site-Specific Transgene Insertion in Primary Human Monocytes

As site-specific insertion of transgenes in primary human monocytes has not been demonstrated previously, we investigated the feasibility of CRISPR-Cas9 mediated homology directed repair (HDR) for precision transgene insertion. We, and others [55,56], have found that transfection of plasmid-based HDR templates causes toxicity in primary immune cells; therefore, we tested recombinant adeno-associated virus (rAAV) for DNA donor template delivery. As rAAV pseudotyped 6 (rAAV6) has been reported to efficiently transduce several different types of primary human immune cells, including myeloid cells [28,32,57], we tested whether rAAV6 is suitable for delivering a DNA donor template for CRISPR-Cas9 mediated HDR. To find optimal transduction efficiency, we utilized rAAV6 encoding a *EGFP* expression cassette under the regulation of the MND promoter, a strong synthetic promoter [58] (Figure 6D). The MND promoter drives transient expression of GFP allowing us to quantify transduction efficiency of rAAV6 in monocytes across a range of MOIs via flow cytometry (Figure 6E). We observed GFP+ cells at all MOI tested (Figure 6F), with the frequency of GFP+ cells increasing in a dose-dependent manner and plateauing at 5 × 10^5^ (Figure 6E,F). Thus, moving forward we used rAAV6 at 5 × 10^5^ MOI for the subsequent gene knock-in experiments.

Next, we designed a promoter-less, splice-acceptor *EGFP* donor template that targets the *AAVS1* safe harbor locus (Figure 6D). We chose the *AAVS1* as it has no known biological function in mammalian cells and has been coined as a “safe harbor” for transgene expression [59]. Moreover, successful integration is required for *EGFP* expression under the regulation of the endogenous *AAVS1* promoter, eliminating any possible episomal vector expression. In our previous report using this approach in primary human B cells, we observed high rates of integration (>60%) [60]. Surprisingly, only 5% of the monocytes were observed to be successfully engineered as measured by flow cytometry (Figure 6E,F). These results indicate that CRISPR-Cas9 can mediate HDR for transgene insertion in monocytes, but at very low frequency.

### 2.7. Genetically Engineered Primary Human Monocytes Have Enhanced Anti-Cancer Activity In Vitro

Finally, we examined whether primary human monocytes can be genetically modified to increase anti-cancer activity. Previous work by Weissman and colleagues reported that antibody blockade of SIRPα/CD47 interaction in conjunction with rituximab (anti-CD20) allows for enhanced antibody-dependent cellular phagocytosis (ADCP) and antibody-dependent cellular cytotoxicity (ADCC) of macrophages [38,39,61]. Therefore, we tested whether CRISPR-Cas9 mediated *SIRPA* gene knockout in monocytes can enhance ADCP and ADCC against a rituximab-opsonized CD47-expressing cancer cell line, i.e., Raji cells (Figure 7A). This approach has the potential to be superior to the use of blocking antibodies due to reduced systemic toxicity as the gene edits are autonomous to the engineered monocytes. To this end, monocytes were transfected with CRISPR-Cas9 targeting the *SIRPA* locus, rested for 3 days in a low cytokine media to allow editing to occur, and then further differentiated into M1 macrophage (M1 MDMs) and assessed by flow cytometry to confirm SIRPα protein loss. Interestingly, this revealed that SIRPα protein was highly reduced in knockout samples but not completely abolished when compared to an isotype control (Appendix A), indicating significant retention of SIRPα protein on the surface membrane of MDMs, potentially as a result of slow turnover of SIRPα post-genetic knockout. The SIRPα knockout M1 macrophages were then subjected to ADCP assays with CFSE-labeled Raji target cells, a cancer cell line derived from a Burkitt lymphoma and expressing CD47 and CD20 on their surface. SIRPα knockout MDMs were co-cultured with rituximab-opsonized-CFSE Raji cells at 1:4 of E:T ratio for 4 h and flow cytometry was used to measure the frequency of phagocytic MDMs. We observed a trend of increasing phagocytic activity in the SIRPα knockout MDMs when compared to wild-type MDMs (13% and 8%, respectively), though it was not statistically significant (Figure 7B,C) despite having a high frequency of SIRPα knockout (Figure 7D,E).

As ADCC activity has been described as one of the core functions of macrophages [19,62], we tested whether SIRPα knockout could increase ADCC in MDMs. To test this, SIRPα knockout and wild-type MDMs were co-cultured with rituximab-opsonized luciferase-expressing Raji cells and luciferase activity was measured at the indicated timepoints (Figure 7F). We observed significantly reduced luciferase activity after 20 h of co-culture in SIRPα knockout MDMs compared to the non-engineered MDMs (Figure 7F), demonstrating that SIRPα knockout enhances ADCC in MDMs.

## 3. Discussion

Much of the knowledge of human monocytes and their biological functions come from carefully compared genomic analysis of sorted monocyte subsets in conjunction with comparison to murine monocytes or monocyte cell lines [63]. Despite the presumed conserved biology of human monocytes, murine monocytes and monocyte cell lines, there are important differences in gene expression, signaling pathways, and cytokine profiles between these cell types [64,65,66]. CRISPR-Cas9 gene editing has opened new avenues to study primary human monocytes and their downstream effectors via direct modification of their genomes [34]. In addition, it enables the ability to create specifically engineered monocytes for the potential treatment of various diseases and disorders. Here, we report on a systematic optimization of a simple and flexible CRISPR-Cas9-based engineering platform that can be used in human monocytes. Our findings that monocytes can be efficiently edited using CRISPR-Cas9 RNP agree with a recent manuscript from Hiatt et al. on the generation of isogenic primary human myeloid cells using CRISPR-Cas9 RNPs [34].

In addition, we closely examined deploying Cas9 mRNA in place of Cas9 protein to enable the use of rapidly evolving engineering tools, such as base and prime editors where purified proteins are not commercially available or difficult to produce [67]. Interestingly, in stark contrast to our finding in other immune cell types, such as T and B cells [27,31,68], Cas9 mRNA yielded inefficient and highly inconsistent genome editing across multiple independent monocyte donors and genomic target sites. By analyzing data from the BLUPRINT project, we discovered that some RNases are expressed at high levels in monocytes compared to T cells (i.e., RNase 2 and 6). In addition, RNase activity assay indicates inconsistent but high levels of RNase activity among monocytes donors, while consistently low levels of RNase activity observed in T cell lysate. These led us to deploy a pan-RNase inhibitor to reduce RNA degradation, which rescued Cas9 mRNA-based gene editing efficacy. Altogether, these observations indicate that members of RNase A superfamily in monocytes may be responsible for inefficient and inconsistent mRNA-based genome editing. Similarly, Peterson and Venkataraman et al. (2021) demonstrated that intracellular RNases were responsible for poor mRNA expression and introducing RNase inhibitor along with mRNA-based engineering platforms allowed for higher engineering efficiencies in non-human primate hematopoietic stem cells [48]. In addition, we demonstrate the use of a mRNA encoding base editor for multiplex genome editing, allowing for safe, multiplex genome modification without risks associated with DSBs, i.e., DNA damage response, chromosomal loss, or translocations. Although the use of iRNase significantly improves engineering efficiencies when deploying mRNAs in monocytes, the effects of transient iRNase on monocytes phenotype and function has not been addressed in this study. Thus, future experiments are needed to address this potential issue.

Beyond gene knockout, the ability to stably express heterologous transgenes in primary human monocytes is an attractive aspect for both basic research and therapeutic applications. To this end, we demonstrated that the CRISPR-Cas9 system can be combined with rAAV DNA donor delivery to mediate precise integration of a transgene through HDR in primary human monocytes, albeit only at ~5% integration efficiency. As homologous recombination occurs largely during the S to G2 phases of the cell cycle [69], this may be explained by the fact that monocytes are largely non-dividing in vitro. Yao et al. demonstrated that CRISPR-Cas9 combined with linearized double-stranded DNA template mediates homology-mediated end joining (HMEJ) or microhomology-mediated end joining (MMEJ) can improve precise integration of a transgene in non-dividing cells [70]. However, we observed that electroporating a donor DNA template into monocytes is highly toxic and leads to complete loss of transfected cells. Alternative routes to deliver DNA donor template and HDR-independent integration approaches are thus needed to optimize the efficiency of CRISPR-Cas9 mediated site-specific transgene integration in human monocytes. A previous study reported that dual *MafB/c-Maf* knockout mice produce functional mouse macrophages with highly enhanced self-renewal potential [71]. Provocatively, it may be possible to transiently knockdown, or permanently knockout, *MAFB/C-MAF* in human monocytes to establish a proliferating primary human monocyte amenable to efficient HDR for transgene integration. However, there are situations where stable integration is not strictly required. For instance, Klichinsky et. al., demonstrated the use of a replication-incompetent adenoviral vector (Ad5f35) for transient expression of a chimeric antigen receptor (CAR) in macrophages and showed long lasting transgene expression in mice (up to 100 days) [72].

ADCP and ADCC are critical mechanisms of anti-tumor activity in myeloid cells [38,39]. SIRPα is a transmembrane protein that regulates ADCP and ADCC in innate immune cells by interacting with the broadly expressed CD47 molecule, conferring a “don’t eat me” signal that prevents phagocytosis of healthy cells [61,73]. Cancer cells exploit this negative regulatory mechanism by overexpressing CD47 on their surface to escape immune surveillance and blockade of SIRPα/CD47 interaction is reported to enhance anti-tumor ADCP and ADCC [36,38,39]. In an attempt to enhance ADCP and ADCC through genetic engineering, we used CRISPR-Cas9 to knockout SIRPα in primary human monocytes. Despite a high frequency of protein reduction in the monocyte population, we observed no significant difference in ADCP between experimental and control groups. In contrast, we observed significantly enhanced ADCC in SIRPα knockout MDMs compared to non-edited MDMs.

Interestingly, flow cytometry data revealed extensive persistence of low level of SIPRα protein despite high rates of genetic knockout, presumably due to low protein turnover in the non-activated, quiescent state of MDMs. This slow turnover of SIRPα protein in MDMs may have reduced the expected enhanced efficiency of ADCP, suggesting that ADCP may be more sensitive to the SIRPα protein retention than the ADCC. In line with this, Wettersten et. al., reported that macrophage were able to kill CD47-expressing lung cancer when opsonized with anti-integrin b-3 antibody through ADCC mechanism instead of ADCP [74]. Since ADCC activity increased with opsonized target cells without SIRPα/CD47 blockade, it suggests ADCC may not be sensitive to the ‘don’t eat me’ signal, however, reducing SIRPa/CD47 signaling intensity somehow augmented the ADCC activity. Further investigation needs to be carried out to explain this phenomenon. To further address the potential protein retention issues, future work could knockout *SIRPA* in a developmental-like model. For example, Dubois and Craft et al. demonstrated that *SIRPA*, a hallmark of cardiomyocyte marker, can be upregulated during differentiation of iPSCs into cardiomyocytes [75]. Therefore, knocking out *SIRPA* in iPSC before differentiating them into M1 macrophage may address the hypothetical protein retention issue.

In summary, we have developed methods for genome engineering of primary human monocytes using either protein or mRNA-based engineering reagents and demonstrate that base editor can be used for efficient multiplex gene knockout in primary human monocytes. While we reported high transduction efficiency using rAAV6, HDR with rAAV6 DNA-template donor and CRISPR-Cas9 reagents mediated only low EGFP knock-in efficiency. Future studies are thus required to identify methods that permit high efficiency site-specific transgene integration in monocytes to further bolster the ability to study monocyte biology and the development of engineered monocyte based cellular therapies.

## 4. Materials and Methods

### 4.1. Monocyte Isolation and Culture

Human leukapheresis samples from de-identified, normal, healthy donors were obtained by automated leukapheresis (Memorial Blood Centers, Minneapolis, MN, USA) following informed consent with approval from the University of Minnesota Institutional Review Board (IRB 1602E84302). Leukapheresis samples were processed with a Ficoll-Paque (Cytiva, Marlborough, MA, USA) density gradient in accordance with the manufacturer’s instructions. Red blood cells were then lysed with ACK buffer (Quality Biological, Gaithersburg, MD, USA) and CD14^+^ monocytes were then isolated by negative selection using an EasySep Human Monocyte Isolation Kit (STEMCELL Technologies, Vancouver, Canada).

Sorted Monocytes were cultured at 5 × 10^5^ cells/mL at 37C in 5% CO_2_ on ultra-low attachment plates (Corning Costar, Corning, NY, USA) in RPMI supplemented with 10% heat-inactivated FBS (R&D system, Minneapolis, MN, USA), 1% Penicillin and Streptomycin (P/S) (Invitrogen, Waltham, MA, USA), and 5 ng/mL of either recombinant human granulocyte-macrophage colony-stimulating factor (rhGM-CSF; R&D Systems, Minneapolis, MN, USA) or recombinant human macrophage colony-stimulating factor (rhM-CSF; R&D Systems, Minneapolis, MN, USA). Media was refreshed every 3–4 days throughout the duration of all experiments unless stated otherwise.

### 4.2. M1 Polarization

Monocytes were cultured in RPMI supplemented 10% FBS, 1% P/S, and 25 ng/mL GM-CSF (R&D Systems) for 7 days followed by the addition of 100 ng/ml IFN-γ (PeproTech, Cranbury, NJ, USA) and 200 ng/ml LPS (Sigma-Aldrich, St. Louis, MO, USA) for an additional 24 h.

### 4.3. Guide RNA Design and Cas9 RNP Preparation

Single guide RNA (sgRNA) with SpCas9 targeting *AAVS1*, *CCR5*, *CD14*, *PDCD1*, *TP53*, *SIRPα*, *β2M*, or *PD-L1* (Appendix A) were designed using online tools (Benchling or Synthego). sgRNA were reconstituted with DNase/RNase/protease free 1× TE buffer (Fisher Scientific, Waltham, MA) at 1 mg/mL. sgRNAs with cytidine base editor (CBE) targeting *β2M* or *PDL-1* were designed to target a splice donor site using SpliceR^20^ (http://z.umn.edu/spliceR, accessed on 9 June 2020). All sgRNAs were synthesized with chemically modified 2′-O-methyl and 3′ phosphorothioate internucleotide modified linkages at the 3′ and 5′ ends (IDT, Coralville, IA), as previously described [76].

CRISPR-Cas9 ribonucleoprotein (RNP) complexes were formed by combining 1 μg respective sgRNAs with 5 μg SpCas9 V3 protein (Alt-R^®^, IDT, Coralville, IA, USA) and incubating at room temperature (RT) for a minimum of 20 min before use.

### 4.4. Monocyte Electroporation

Monocytes were plated in RPMI, supplemented with 10% heat-inactivated FBS, and 1% P/S at 1 × 10^6^ cells/mL in an ultra-low attachment plate for 3 h prior to transfection. Monocytes were collected and cell pellets were washed 1× PBS followed by resuspension in T buffer (Fisher Scientific, Waltham, MA, USA) at a concentration of 5 × 10^7^ cells/mL. Cells were then loaded into a 10 uL Neon pipette tip and electroporated using a Neon Transfection System (Fisher Scientific, Waltham, MA, USA) with E5 protocol (1700 volts, 20 millisecond bandwidths, 2 pulses) unless indicated otherwise. Electroporated monocytes were then immediately transferred to 24-well plates containing 1 mL of complete monocyte medium at 5 × 10^5^ cells/ml and placed in the tissue culture incubator.

A similar procedure was followed for experiments using chemically modified Cas9 (Cas9) mRNA instead of Cas9 RNPs. In these experiments 1 μg of sgRNA was combined with either 1.5 μg SpCas9 mRNA (CleanCap^®^, Trilink Biotechnologies, San Diego, CA, USA) or 1.5 μg coBE4 mRNA (Trilink Biotechnologies, San Diego, CA, USA) prior to addition to cells.

### 4.5. Quantification of Genome Editing

Genomic analysis was performed following DNA extraction of 2 × 10^5^ cells using a standard DNA extraction kit (Fisher Scientific, Hampton, NH, USA) followed by Accuprime high fidelity taq (Invitrogen, Waltham, MA, USA) PCR amplification of the gene of interest using PCR primers flanking the sgRNA target site (Appendix A). Amplicons from the PCR reaction were purified using a PCR purification kit (Qiagen, Hilden, Germany) and then submitted for Sanger sequencing (Eurofins Genomics, Louisville, KY, USA). Sequencing chromatograms from SpCas9 mediated gene editing were analyzed with their respective non-edited control chromatograms using online tools (https://tide.deskgen.com, accessed on 10 October 2021 or https://ice.synthego.com, accessed on 18 May 2022) to quantify insertions or deletions (indels). Sequencing chromatograms from coBE4 mediated base editing were analyzed using the EditR online tool [54] (z.umn.edu/editr accessed on 30 April 2020) to quantify percent C to T conversion (% C > T).

### 4.6. Flow Cytometry

Monocytes and MDMs were washed twice in Gibco 1× PBS (Fisher Scientific, Waltham, MA, USA) supplemented with 0.5% BSA (Sigma-Aldrich, St. Louis, MO, USA), 2 mM EDTA (Invitrogen, Waltham, MA, USA), incubated with human TruStain FcX (Fc Receptor Blocking solution (Biolegend, San Diego, CA, USA) for 5 min at room temperature and incubated with Fixable Viability Dye eBioscience eFluor780 (Fisher Scientific, Waltham, MA, USA) for an additional 10 min at room temperature. Cells were then washed with Flow Buffer and labeled with fluorescent-conjugated antibodies (Appendix A) for 15 min at room temperature. Cells were then washed once, resuspended in Flow Buffer, and analyzed on an LSR II (BD Biosciences, Mississauga, Canada), Fortessa (BD Biosciences, Mississauga, Canada), or CytoFlex S (Beckman Coulter, Brea, CA, USA) flow cytometer. Data analysis was performed with FlowJo version 10.7.1 software (FlowJo, Ashland, OR, USA).

### 4.7. RNase Alert Assay

An RNase Alert Kit (IDT, Coralview, IA, USA) was used to compare RNase activity in 3 independent donors of monocytes and T cells. Monocytes were rested for 3 h, and T cells were stimulated for 48 h [27] prior subjected to RNase Alert Assay. Cells at a total of 5 × 10^5^ cells were resuspended in a 50 mL 1× PBS and subsequent homogenization using a FisherBrand Sonicator model FB50 with a probe model CL-18 (Fisher Scientific) with settings of 30 Amplitude for 10 s at a total of 3 intervals (on ice). Then, the tissue homogenates were centrifuge at 12,000× *g* for 25 min at 4C to obtain tissue lysates. The tissue lysates were mixed with FAM-conjugated RNase Alert substrate following manufacturer’s protocol and incubated at 37C for one hour. RNase A (provided in the kit) was used as positive control. Water and PBS were used as negative controls. After one hour incubation, the reaction tubes were captured under the UV light using a Bio-Rad gel imager and subsequent a single read of the FAM fluorescent signal using a BioTek Synergy microplate reader with Gen 5 program version 2.04 at excitation of 495 nm and emission of 520 nm.

### 4.8. rAAV Design and Transduction

To measure transduction efficiency of rAAV6, an expression cassette encoding of *EGFP* under an MND promoter was synthesized (Integrated DNA Technologies, Coralview, IA, USA), cloned onto the rAAV backbone, and sent to package into rAVV6 capsid at a reputable commercial facility (Vigene Biosciences, Rockville, MD, USA). This vector was then transduced into monocytes at multiple of infections (MOIs) of 0.1, 0.2, 0.5, or 1 × 10^6^ VP/cell and cells were then cultured for 7 days. with the media replaced every 48 h.

To determine transgene insertion efficiency, an expression cassette encoding a splice acceptor followed by *EGFP* cDNA, a BGH polyA sequence and flanked by 1000 base-pair homology arms targeting human *AAVS1* locus was synthesized (Integrated DNA Technologies, Coralview, IA, USA), cloned into an rAAV backbone, and sent to package into rAAV6 viral capsid at a reputable commercial facility (Vigene Biosciences, Rockville, MD, USA). This donor template was transduced into monocytes at a MOI of 5 × 10^5^ VP/cell immediately following electroporation with Cas9 RNPs targeting the *AAVS1* locus in monocytes. Cells were then cultured for 7 days. with the media replaced every 48 h.

### 4.9. Quantification of Phagocytosis

M1 MDMs were washed twice with 1× PBS and plated at 1 × 10^5^ cells/mL/well on a 24-well non-coated tissue culture plate for 24 h in RPMI supplemented 10% FBS, 1% P/S, and 25 ng/mL GM-CSF. After 24 h, cells were washed twice with 1× PBS without dislodging, and resuspended in RPMI without FBS. Simultaneously, target Raji cells were labeled with CFSE (Invitrogen, Walthman, MA, USA) in accordance with the manufacturer’s protocols and opsonized by incubation with 10ug/mL Rituximab (Genetech, South San Francisco, CA, USA) for 30 min at room temperature. Opsonized, CFSE-labeled Raji cells were then co-cultured with MDMs at 1:4 E:T ratio at 37 °C and 5% CO_2_ for 4 h. Cells were then washed twice with 1× PBS and stained with viability dye and a fluorescently labeled antibody against CD86 (Biolegend, San Diego, CA, USA). Samples were then analyzed on a CytoFlex S flow cytometer.

### 4.10. Quantification of Antibody-Dependent Cellular Cytoxicity (ADCC)

ADCC was performed as previously described [62] to determine functional enhancement of engineered MDMs. Briefly, MDMs were co-cultured with luciferase-expressing Raji cells at an E:T ratio of 3:1 and luminescent intensity was measured after 0.5, 2 and 20 h of co-culture. Percent relative luciferase activity was calculated by Relative luminescent unit (RLU) of MDM + Raji divided by RLU of Raji only and multiplied by 100.

### 4.11. Statistical Analysis and Figure Construction

The Welch’s *t*-test was used to evaluate significant differences between two groups. Differences between three or more groups with one data point were evaluated by a one-way ANOVA test. Differences between three or more groups with multiple data points were evaluated by a two-way ANOVA test. All assays were repeated in at least three independent donors. Means values ± SD are shown. The levels of significance were set at *p* < 0.05. All statistical analyses were performed using GraphPad Prism 9.2.0 (GraphPad, San Diego, CA, USA). Schematics were constructed using Biorender.com (Toronto, ON, USA).

## 5. Patents

Genome Engineering Primary Human Monocytes. PCT/US2019/037986.

## Figures and Tables

**Figure 1 ijms-23-09749-f001:**
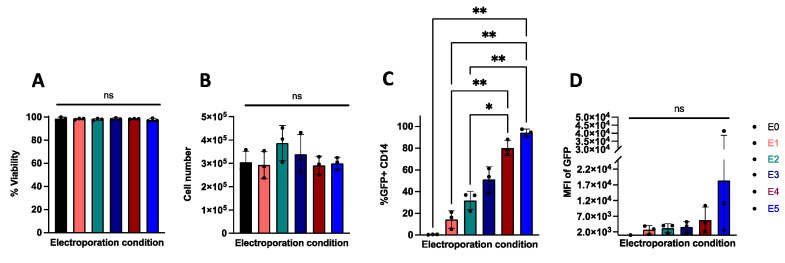
Electroporation protocols for transfection efficiency, viability, and cell recovery of primary human monocytes. (**A**) Percent viable, (**B**) cell count, (**C**) and percent GFP+CD14+ of total cells as well as (**D**) GFP MFI of GFP+ cells following electroporation with EGFP mRNA with the indicated electroporation conditions. Statistical analyses were performed using one-way ANOVA followed by Dunnett’s T3 multiple comparison test (n = 3 independent biological donors) (ns = *p* > 0.05, * *p* < 0.05, ** *p* < 0.01).

**Figure 2 ijms-23-09749-f002:**
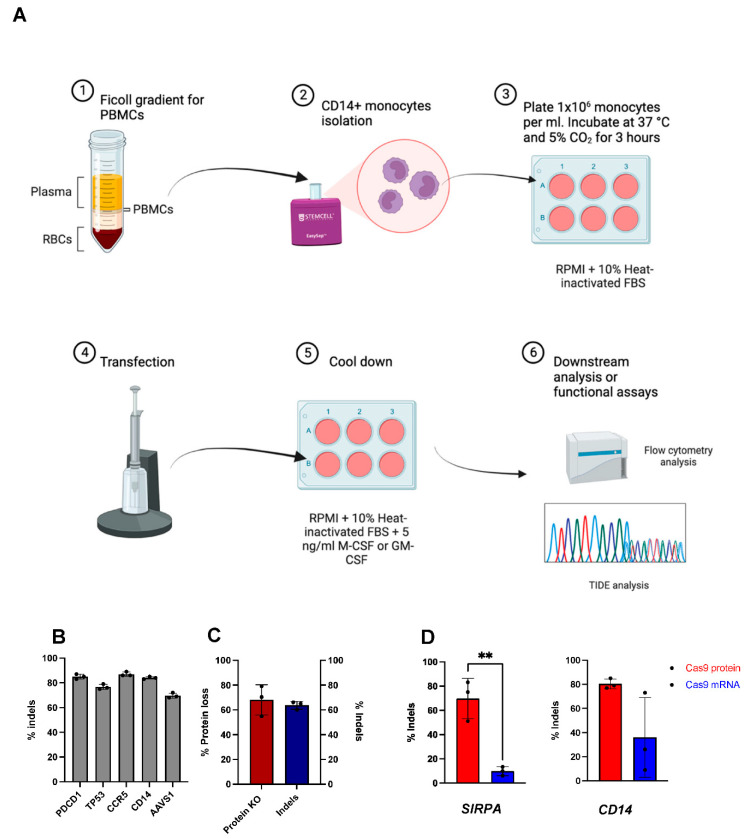
CRISPR-Cas9 mediates genome editing in primary human monocytes. (**A**) Schematic depicting monocyte editing workflow. Monocytes are isolated from PBMCs of healthy human donors and then electroporated with GFP mRNA or Cas9 and sgRNA targeting gene of interest. Electroporated cells are then analyzed for gene editing and protein loss. (**B**) Percent indel formation of gene targeted for CRISPR-Cas9 RNP editing as measured by TIDE. (**C**) Bar graphs showing levels of protein loss (left bar) and indel formation (right bar) (*p* = 0.6735) post Cas9 RNP targeted *SIRPa* KO. (**D**) Bar graph depicting the level of indel formation at *SIRPA* and *CD14* after treatment with Cas9 RNP or Cas9 mRNA and either *SIPRA* or *CD14* sgRNA (n = 3 independent biological donors) (** *p* < 0.01). Statistical analyses were conducted using unpaired Welch’s t test.

**Figure 3 ijms-23-09749-f003:**
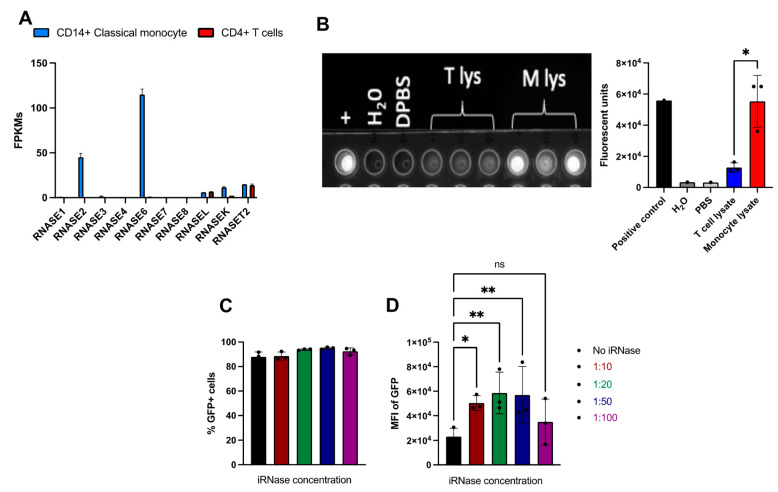
RNase expression profiles in monocytes and T cells, and RNase inhibitor (iRNase) enhanced MFIs of GFP in monocytes. (**A**) mRNA expression profiles of RNases in monocytes and T cells from BLUEPRINT project (ebi.ac.uk). (**B**) Top view picture of the RNase assay depicts qualitative measurement (left panel) and bar graph depicts quantitative measurement (right panel) of RNase activity in T-cell lysate (T lys), monocyte lysate (M lys). (**C**) Bar graph showed % GFP-positive cells, and (**D**) MFI of GFP+ cells 24 h after electroporation of GFP mRNA along with iRNase at the indicated concentrations. Statistical analyses were performed using two-way ANOVA followed by Dunnett’s T3 multiple comparison test (n = 3 independent biological donors) (ns = *p* > 0.05, * *p* < 0.05, ** *p* < 0.01).

**Figure 4 ijms-23-09749-f004:**
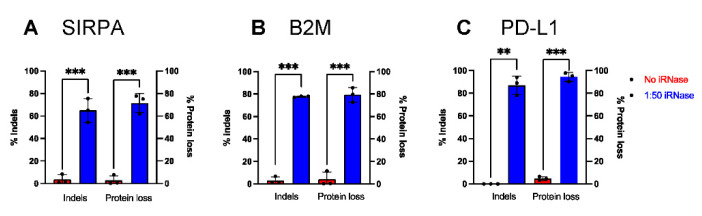
iRNase enhanced Cas9 mRNA mediates various gene KO in monocytes. Percent of monocytes with indels, as measured by TIDE analysis, or loss of protein expression, as measured by flow cytometer following electroporation with Cas9 and gRNAs targeting either (**A**) *SIRPA* (**B**) *B2M* and (**C**) *PD-L1* in the presence (Blue) or absence of iRNase (Red). Statistical analyses were performed using Student’s t-test (n = 3 independent biological donors) (** *p* < 0.01, *** *p* < 0.001).

**Figure 5 ijms-23-09749-f005:**
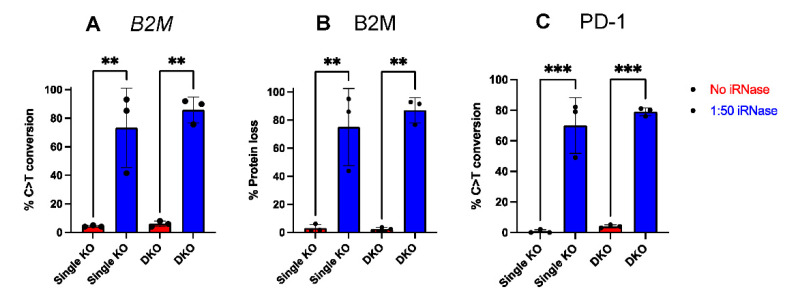
iRNase allows Base editor mRNA for efficient single or multiple genes KO in monocytes. Percent of (**A**) B2M protein loss as measured by flow cytometry, (**B**) percent of C > T conversion at the B2M locus, (**C**) and percent of C > T conversion at the PD1 locus in monocytes following editing with Base Editor mRNA with gRNA(s) target either B2M or PD1 single KO, or double KO along with or without iRNase. Statistical analyses were performed using Student’s *t*-test (n = 3 independent biological donors) (** *p* < 0.01, *** *p* < 0.001).

**Figure 6 ijms-23-09749-f006:**
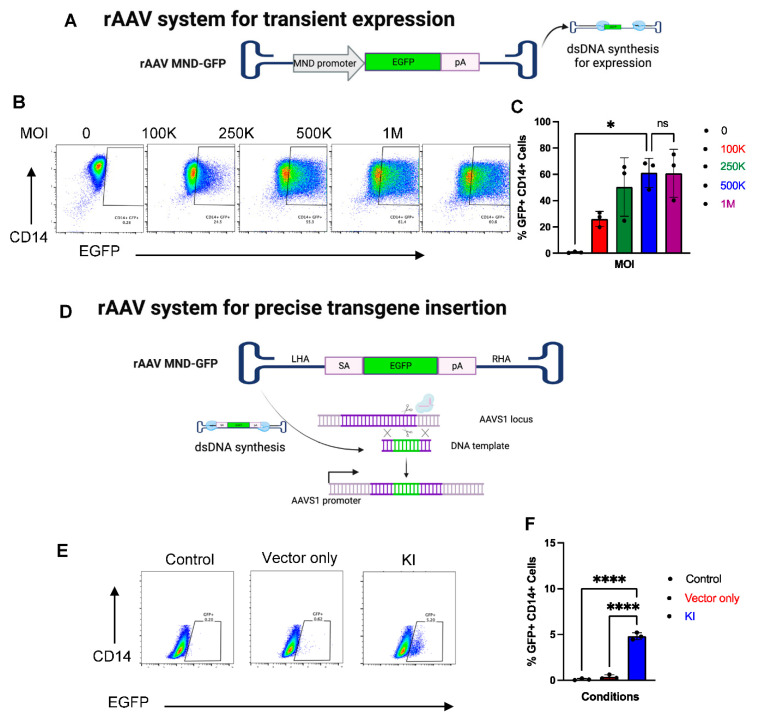
CIRSPR/Cas9 mediate *EGFP* knock-in in monocytes. (**A**) rAAV6 *MND-EGFP* was used for transient expression of *EGFP*. (**B**) Representative flow plots and (**C**) bar graphs showing the frequency of EGFP+ monocytes following transient transduction with AAV encoding MND-EGFP at the indicated MOIs. (**D**) rAAV6 *SA-GFP* was used as a donor DNA template for CRISPR-Cas9 mediated site-specific insertion of *EGFP* at AAVS1 locus. (**E**) Representative flow plots and (**F**) bar graphs showing the frequency of EGFP+ monocytes following post CRISPR-Cas9 engineering with rAAV6 SA-*EGFP* mediated site-specific insertion of *EGFP* at AAVS1 locus. Statistical analyses were performed using one-way ANOVA followed by Dunnett’s multiple comparison test (n = 3 independent biological donors) (ns = *p* > 0.05, * *p* < 0.05, **** *p* < 0.0001).

**Figure 7 ijms-23-09749-f007:**
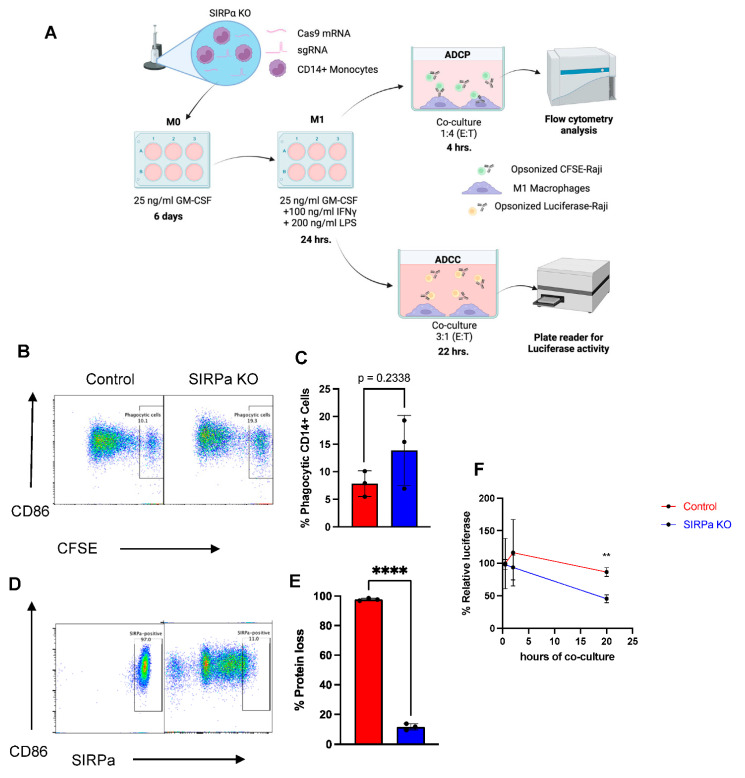
SIRPa KO in monocytes to enhance anti-tumor activity. (**A**) Schematic depicts experimental timeline for ADCP and ADCC. (**B**) Representative flow plots and (**C**) bar graph depicting percentage of phagocytosis as measured by CFSE dye follow coculture of monocytes with CFSE-labeled target cells. (**D**) Representative flow plot depicting and (**E**) bar graph showing frequency SIRPa in M1 macrophages following editing with Cas9 and *SIRPA* gRNA. (**F**) Percentage relative luciferase activity in cocultures of SIRPa KO monocytes or unedited controls with opsonized, luciferase-labeled controls. Statistical analysis was performed using either Student’s t test or two-way ANOVA followed by Sidak’s multiple comparisons test (n = 3 independent biological donors) (** *p* < 0.01, **** *p* < 0.0001).

**Table 1 ijms-23-09749-t001:** Electroporation conditions. Voltage, bandwidth, and pulse settings tested for electroporation of primary human monocytes.

Protocol Name	Voltage (Volts)	Bandwidth (Milliseconds)	Number of Pulses
E0	0	0	0
E1	1300	10	3
E2	1400	10	3
E3	1500	30	1
E4	1600	10	3
E5	1700	20	2

## Data Availability

The data that support the findings of this study are available from the corresponding author, BSM, upon request.

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
