# Peer review of "A Pan-RNase Inhibitor Enabling CRISPR-mRNA Platforms for Engineering of Primary Human Monocytes"

_ijms, 2022, doi:10.3390/ijms23179749_

Round 1
Reviewer 1 Report
In this manuscript, Laoharawee et al reported high-efficient genome engineering in primary human monocytes. The authors found that CRISPR-Cas9 RNPs can be used for efficient gene knockout and inhibition of endogenous RNase activities allows efficient genome editing with mRNA-based delivery of Cas9 and base editor enzymes in primary human monocytes. Using these optimized methods, the authors were able to achieve site-specific gene knock in and specific knock out of the SIRPa gene in primary human monocytes.
Minor concerns:
11) In figure 2A, step 6 is missing.
Author Response
Dear Reviewer 1,
Thank you so much for your time and a wonderful summary of our manuscript. We appreciate it. Please see our response below in red texts.
Yours,
Kanut
In this manuscript, Laoharawee et al reported high-efficient genome engineering in primary human monocytes. The authors found that CRISPR-Cas9 RNPs can be used for efficient gene knockout and inhibition of endogenous RNase activities allows efficient genome editing with mRNA-based delivery of Cas9 and base editor enzymes in primary human monocytes. Using these optimized methods, the authors were able to achieve site-specific gene knock in and specific knock out of the SIRPa gene in primary human monocytes.
Minor concerns:
11) In figure 2A, step 6 is missing. Very good catch! Thank you so much. I corrected the image as suggested.

Reviewer 2 Report
The authors describe a very detailed and highly informative research article and cover several important new procedures in the field of monocyte genome editing: (A) Improvement of THP1 and primary monocyte gene editing using mRNA based editors, enhanced by the use of a pan-RNase inhibitor (iRNase). (B) Successful application of their technical improvement by using AAV6 derived donors for primary monocyte HDR. (C) Demonstration of functionality of a SIRPa KO in primary human monocytes. The MS is of a very high quality, probably supported by the previously filed patent application for the described technique. The technology and all applications are new and have not been previously presented in this format. The discussion is very detailed and recent quotations in a similar context are considered and well incorporated.
I have no fundamental issues with the current MS and am very much in favor of accepting it after only few minor suggestions have been considered.
Minor concerns/suggestions:
Line 91: Please indicate in the text which machine has been used (LifeTech Neon), to allow the reader immediate understanding of the parameters you changed.
· Line 107: please state the chemical modification of the sygRNA in the text (I found it in the M&M, however, it is easier to have it available in situ).
· Line 137: Please add additional information (also in M&M) which SIRP antibodies were used for negative detection in your FACS assay.
· Line 176: You could add the concentration of iRNase used in your experimental setups in the body of the text. Being one of your key messages this will allow readers to gain immediate access to the data in their context. You mention a final 1:50 dilution; maybe the final concentration would be more accurate (if possible)?
· Line 293: “Only 5%”. This is not an entirely bad value, to be honest… ;-). Not great, but you can work well with that level...
· Line 460: As a note on side: the CRISPR MIT algorithm is no longer available (violation of repeat structure considerations) and as such should be removed from your list.
Author Response
Dear Reviewer 2,
Thank you for your time, a wonderful summary of my manuscript, and suggestions to improve the clarity of my manuscript. Please see our responses too your comments in red text below. Please let me know if you have any questions.
Yours,
Kanut
The authors describe a very detailed and highly informative research article and cover several important new procedures in the field of monocyte genome editing: (A) Improvement of THP1 and primary monocyte gene editing using mRNA based editors, enhanced by the use of a pan-RNase inhibitor (iRNase). (B) Successful application of their technical improvement by using AAV6 derived donors for primary monocyte HDR. (C) Demonstration of functionality of a SIRPa KO in primary human monocytes. The MS is of a very high quality, probably supported by the previously filed patent application for the described technique. The technology and all applications are new and have not been previously presented in this format. The discussion is very detailed and recent quotations in a similar context are considered and well incorporated.
I have no fundamental issues with the current MS and am very much in favor of accepting it after only few minor suggestions have been considered.
Minor concerns/suggestions:
Line 91: Please indicate in the text which machine has been used (LifeTech Neon), to allow the reader immediate understanding of the parameters you changed. Thank you for your suggestion. I added Neon in the text as suggested.
- Line 107: please state the chemical modification of the sygRNA in the text (I found it in the M&M, however, it is easier to have it available in situ). Thank you for your suggestion. I added “chemically modified” in the text as suggested.
- Line 137: Please add additional information (also in M&M) which SIRP antibodies were used for negative detection in your FACS assay. Thank you for your suggestion. I added the additional information as suggested.
- Line 176: You could add the concentration of iRNase used in your experimental setups in the body of the text. Being one of your key messages this will allow readers to gain immediate access to the data in their context. You mention a final 1:50 dilution; maybe the final concentration would be more accurate (if possible)? Thank you for your suggestion. I added the different concentration of iRNase used in the experimental setups and added the final concentration (10.8 Units) in the text as suggested.
- Line 293: “Only 5%”. This is not an entirely bad value, to be honest… ;-). Not great, but you can work well with that level... Thank you! I’m happy to hear that this 5% KI value can be useful.
- Line 460: As a note on side: the CRISPR MIT algorithm is no longer available (violation of repeat structure considerations) and as such should be removed from your list. I removed the MIT from the text as suggested.

Reviewer 3 Report
In this manuscript, Laoharawee et al reported a new method to improve the efficiency to genetically modify the human monocytes. Specifically, the authors show that the ineffienciet genome editing by Cas9 mRNA could be mediated by the low RNA inhibitor activity. They show that by co-electroporating RNAase inhibitor the genome editing efficiency can be improved. They also showed prove of principle of the use of rAAV system for the generation of tansgene. The optimisation of the system can be useful for the field. However several aspects need to be addressed to fully validate the results shown in the masucirpt.
Specific points are as below:
1) the inhibition of RNAase activity can significantly interfere the RNA homeostasis. Would this affect the function or fitness of the T-cell for long -term ? This is an important point and need to be addressed further.
2) the GFP mRNA electroporation efficiency under various electroporation conditions are not clearly shown: the three replicates in E5 clearly have much higher SD than other conditions. More replicates are needed to demonstrate that E5 condition is the best amongst all conditions.
3) it seemed the 5 conditions that the authors tested show a progressive increase in their efficiency. Have the authors tried to broader the Bandwidth and number of pulses to see whether other conditions can be better than the current ones?
4) the results from other systems suggest that Cas9 coding sequence is rather too long to be effectively delivered by electrporation or AAV system. The use of short version of Cas9 has been recently reported. Have the author considered to electorate the short Cas9 version and compare the efficiency with with their current protocols?
Author Response
Dear Reviewer 3,
Thank you for your time and a wonderful summary of our manuscript. Your make fantastic points to improve monocyte engineering research. We appreciate it. Please see our responses to your points in red text. Please let us know if you have any questions.
Yours,
Kanut
In this manuscript, Laoharawee et al reported a new method to improve the efficiency to genetically modify the human monocytes. Specifically, the authors show that the ineffienciet genome editing by Cas9 mRNA could be mediated by the low RNA inhibitor activity. They show that by co-electroporating RNAase inhibitor the genome editing efficiency can be improved. They also showed prove of principle of the use of rAAV system for the generation of tansgene. The optimisation of the system can be useful for the field. However several aspects need to be addressed to fully validate the results shown in the masucirpt.
Specific points are as below:
1) the inhibition of RNAase activity can significantly interfere the RNA homeostasis. Would this affect the function or fitness of the T-cell for long -term ? This is an important point and need to be addressed further. This is a good point. The RNase is used in a very transient fashion but indeed this could influence monocyte biology or function. As such, we have added text in the discussion that future experiment is needed to address this issue (line 576).
2) the GFP mRNA electroporation efficiency under various electroporation conditions are not clearly shown: the three replicates in E5 clearly have much higher SD than other conditions. More replicates are needed to demonstrate that E5 condition is the best amongst all conditions. Thank you for your comment. In figure 1C, we show significantly higher %GFP positive monocytes when using E5 protocol, indicating the highest transfection efficiency of programs tested. In figure 1D, we show that the MFI of EGFP using the E5 protocol was not significantly higher than other programs, which is likely due to the variation of RNase levels across donors (as seen in figure 3B where 1 donor has much lower RNase activity). However, in subsequent experiments that use mRNA-based genome editing reagents in combination with RNase inhibitor do not show this variability across donors and produce very high rates of editing (Figure 4 and 5).
3) it seemed the 5 conditions that the authors tested show a progressive increase in their efficiency. Have the authors tried to broader the Bandwidth and number of pulses to see whether other conditions can be better than the current ones? Thank you for your comment. No, we did not test further because the E5 resulted in a range of 90-98% transfection efficiencies, which we considered adequate.
4) the results from other systems suggest that Cas9 coding sequence is rather too long to be effectively delivered by electroporation or AAV system. The use of short version of Cas9 has been recently reported. Has the author considered to electorate the short Cas9 version and compare the efficiency with with their current protocols? Thank you for your comment on this. This is very interesting point to test in the future experiment whether the short Cas9 mRNA can improve engineering efficiencies. However, we currently see this as being outside the scope of this report that focuses on the use of RNase inhibitor to enhance both spCas9 and base editor activity when using mRNA based delivery.

Round 2
Reviewer 1 Report
This reviewer is satisfied with the revised manuscript.
Reviewer 2 Report
Dear authors,
Thanks for taking the time to add the minor details. I take the opportunity to congratulate you for a high quality MS and am glad it is made available to the scientific community! On a very small side note: Throughout the MS you use both, SIRPA and SIRPa, maybe you can apply one round of find and replace to unify the nomenclature?
Sincerely.
Reviewer 3 Report
The authors have addressed all my comments and in my opinion the manuscript has reached the sufficient level for acceptance.